# Feedstock-Induced Changes in the Physicochemical Characteristics of Biochars Produced from Different Types of Pecan Wastes

Miaomiao Zhang [1,2], Fangren Peng [1,2], Jinping Yu [3] and Zhuangzhuang Liu [3,*]

1    Co-Innovation Center for Sustainable Forestry in Southern China, Nanjing Forestry University, Nanjing 210037, China; zmmaoee@163.com (M.Z.); frpeng@njfu.edu.cn (F.P.)
2    College of Forestry, Nanjing Forestry University, Nanjing 210037, China
3    Jiangsu Key Laboratory for the Research and Utilization of Plant Resources, Institute of Botany, Jiangsu Province and Chinese Academy of Sciences (Nanjing Botanical Garden Mem. Sun Yat-Sen), Nanjing 210014, China; yujinping@cnbg.net
*    Correspondence: zzliu@jib.ac.cn

**Abstract:** Large amounts of residues are generated in pecan cultivation processes. Biochar is an environmentally friendly way to utilize residues but attempts to prepare and apply biochar with pecan residues are rare. In this study, six types of biochars were produced from pecan branches, trunks, roots, nutshells, husks, and leaves under pyrolysis, and their physicochemical properties were compared to assess their application perspective in environmental and agricultural fields. The yields of six pecan biochars were 32.1%–45.9%, with the highest yield for husk biochar (HB) (45.9%). Among the pecan biochars, trunk biochar (TB) and root biochar (RB) had much larger specific surface areas. Branch biochar (BB), TB, and RB presented tubular structures with elliptical pores, while nutshell biochar (NSB), HB, and leaf biochar (LB) appeared flaky or as clustered structures with relatively rougher outer surfaces and irregular pores. The functional group types of pecan biochars were generally similar, but the intensities of the peak near 2900 cm$^{-1}$ in BB were obviously higher than those of the other biochars. RB and LB contained significantly more ash and volatile than those of the other pecan biochars, with the highest fixed carbon content being found in NSB (70.1%). All of the pecan biochars were alkaline (7.90–9.87), and HB, LB, and NSB had significantly higher pH values than those of the other biochars. Elemental analysis indicated that RB, NSB, and LB had higher carbon levels (more than 70%) with lower O/C ratios (no more than 0.2). HB possessed a relatively high content of nitrogen, potassium, magnesium; the phosphorus content was highest in NSB; LB had the highest calcium content. The results of principal component analysis showed that BB, LB, and NSB were clustered in the same quadrant with relatively close relationships. The results of this study can guide the utilization of pecan wastes and their application as biochar in different fields.

**Keywords:** pecan; feedstock; biomass; pyrolysis; properties



## 1. Introduction

Developed in recent years, biomass carbonization technology is an economic and environmentally friendly technology for the utilization of agricultural and forestry wastes. By this technology, biochar (a highly aromatic solid material) can be prepared from the pyrolytic carbonization of biomass under anoxic or anaerobic atmosphere [1,2]. Biochar is generally an alkaline and has a strong capacity to exchange cation capacity. It also contains various organic components and nutrients. Therefore, it is usually applied as a soil conditioner to change the diversity of microbial communities, improve soil properties, and thus promote the growth of crops [3]. Meanwhile, biochar is characterized with a porous structure and a large surface area, which gives it a strong ability to adsorb water and pollutants [4–6]. The rich functional groups in the surface of biochar can not only increase

the hydrophilicity and chemical reaction activity of biochar, but also improve its adsorption and catalytic performance [7–9]. Therefore, biochar is also widely used for the adsorption of pollutants in soil or sewage. Furthermore, because of its high carbon content and stable C-ring structure, it possesses great potential for carbon sequestration [10,11].

The application effect of biochar in different fields is determined by its physicochemical properties, and the types of feedstocks greatly influence the characteristics of biochar. It has been revealed that the chemical constituents of the feedstocks significantly affect the biochar properties, such as the ash content, pH, electrical conductivity (EC), and ultimate composition. The nutrient content of biochar was found to be linearly correlated with that of its feedstock, and the pore characteristics and surface functional groups mainly depended on the organic component of the feedstocks [12–14]. For instance, Yu et al. found that the ash content was higher in herbaceous plant-based biochar than in woody plant-based biochar, while the organic carbon content was the opposite in the two types of plants [15]. Mukome et al. showed that softwood biochars presented a larger surface area when compared with that of hardwood (maple) biochar [12]. The characteristics of biochars prepared from various organs or tissues of the same plant have also been compared. For example, the leaves, trunks, and branches of *Jatropha curcas* were used as materials for the preparation of biochar, and obvious differences in the biochar properties were revealed [16]. For the feedstock type to have an important effect on the properties and the application of biochar, it is essential to clarify the properties of biochars prepared with different raw materials in order to select suitable biochars for application [17,18].

Pecan [*Carya illinoinensis* (Wangenh.) K. Koch] is an important economic tree with many kinds of uses (nut, woody oil, timber, and landscaping, etc.) [19,20]. According to statistics, the world production of pecans in 2021 was 166,000 tons, an increase of 7% over 2020 production, and 35% over the average production of the previous decade. Pecan planting has increased significantly in the last few years in China, with Anhui Province alone having pecan planting areas that exceed 40,000 hm$^2$ (data as of 2021). As the industrialization of pecans increasingly expands, a large amount of waste is produced. The pecan husk is a part of the fruit that cracks and falls off as the nut approaches maturity. It occupies 25%–30% of the overall weight of the pecan fruit, which implies a massive biomass. After the pecan husk falls off, the pecan nut is left. The pecan nut has a hard outer nutshell, which must be broken and stripped in order to obtain the kernel for eating or processing, resulting in a large amount of residual nutshells. Pecan has a well-developed taproot and few lateral roots. The transplant of pecan seedlings will inevitably lead to some mortality. Moreover, the dead pecan plants will be generated due to weather disasters and improper orchard management, such as the ineffective control of pests and diseases. To replant new plants, the dead branches, trunks, and roots need to be removed, which are the common pecan residues in pecan orchard. Regular pruning is also an important measure to ensure pecan trees are high-quality and productive, but it produces a large amount of pruning waste, such as branches and leaves. In addition, large fallen leaves are also generated during the winter season. At present, these wastes are mainly disposed of by conventional techniques, including incineration, landfilling, and disposal, which not only results in significant resource waste but also pollutes the environment [21,22]. Given the great performance of biochar application in agriculture and environment, it is necessary to prepare biochars using the various pecan wastes and to evaluate their application potential.

With respect to the research on pecan biochars, most of the attention has been directed to the ability of pecan shell biochar to adsorb pollutants, whereas little knowledge has been obtained on the characteristics and application performance of biochars made from other types of residues [23,24]. In a previous study by Zhang et al, two types of biochars were made from pecan nutshells and pruned wastes, and their properties were analyzed [25]. Additionally, Liu et al. prepared biochars of pecan branches, leaves, and nutshells under different carbonization temperatures, evaluating the application potential of different biochars based on their properties [26]. Although more attempts are appearing in the study of biochars from different types of pecan wastes, the pyrolysis preparation of biochar

from major wastes such as trunks, roots, and husks, and the comprehensive analysis of the characteristics of biochar from a variety of pecan wastes have not yet been reported. Furthermore, some important properties (types of surface functional groups, the ash, volatile matter and fixed carbon contents, proximate elemental composition, the size distribution of biochar particles, etc.) remain largely unknown in terms of pecan biochars. In view of this, this study was conducted to prepare biochar via a pyrolysis carbonization process using six types of wastes, including pecan branches, trunks, roots, nutshells, husks, and leaves, as feedstocks. The comparative analysis was further conducted on the physical and chemical characteristics of different pecan biochars, which can provide important references for the application of biochars from pecan wastes in different fields.

## 2. Materials and Methods

### 2.1. Pecan Waste Collection

In November of 2022, the pecan branches, trunks, roots, nutshells, husks, and leaves were gathered from the experimental base of pecans of Nanjing Forestry University (Jurong, Jiangsu Province, China) and brought back to the laboratory. Among the plant residues, fallen leaves and husks were collected from the ground in the orchard. Branches, trunks, and roots were removed separately from the whole dead plants and then crushed with a pulverizer. Nut shells were derived from the process of pecan nuts. The different materials were washed three times using pure water to get rid of soil at a constant temperature oven at 65 °C to achieve a constant weight.

### 2.2. Production of Biochar

Six pecan biochars were prepared using similar methods as previously described [26,27]. Then, the iron containers with samples were put in a KSL-1200X muffle furnace (HeFei-Kejing, China) under an anaerobic condition. The heating rate was set as 10 °C/min, and the pecan biomasses were pyrolyzed at the endpoint temperature of 500 °C for 1 h. After pyrolysis was completed and the atmosphere temperature in the muffle furnace reached room temperature, the biochar samples were removed. The biochars were ground through 20 mesh and 100 mesh sieves, and then sealed and stored in self-sealing bags. Biochars with <20 mesh particle size were used for particle size distribution, surface pore structure, surface morphology scanning, and the determination of pH and EC, while biochars with <100 mesh particle size were used for the detection of surface functional groups, proximate analysis, and the estimation of elemental composition. The industrial analysis and determination of the biochar yield, chemical properties (pH and EC), and element contents were performed three times. The biochars made from branches, trunks, roots, nutshells, husks, and leaves were denoted as branch biochar (BB), trunk biochar (TB), root biochar (RB), nutshell biochar (NSB), husk biochar (HB), and leaf biochar (LB), respectively.

### 2.3. The Determination of the Biochar Properties

The following formula was used to calculate biochar yield (%):

$$Biochar\ yield\ (\%) = (W_b/W_o) \times 100$$

where $W_b$ is the weight (g) of the biochar obtained by an electronic analytical balance (FA1004, LIANGPING, Shanghai, China), and $W_o$ represents the weight (g) of the feedstocks.

The particle size distribution of six pecan biochars was measured with a dry sample testing method using a MS2000 laser particle size analyzer (Malvern Instruments, Malvern, UK). According to the methods and equipment described previously [26], the specific surface area (SSA) and the pore structure (total pore volume–TPV, and average pore diameter–APD) of the pecan biochar samples were measured, and the scanning electron microscope (SEM) photos for the biochar morphology were taken at a magnification of 500× with a scale of 100 μm. The surface functional groups in biochar were detected using Fourier transform infrared (FTIR) spectroscopy (VERTEX 80 V, Bruker, Bremen, Germany).

The proximate analysis of six pecan biochar samples was based on the method of the American Society for Testing and Materials (ASTM). A certain quantity of biochar was added to porcelain crucibles (uncovered crucibles) and pyrolyzed for 6 hours at 750 °C in a muffle furnace. The weight ratio of residual material to the total dry matter was used to calculate ash content [2,28]. In order to determine the content of volatile matter (VM), a certain amount of the biochar sample was weighed, placed in porcelain crucibles (covered crucibles), and put in a muffle furnace to be retained at 900 °C for 6 min. According to the weight of the total dry matter (DMW) and the residual material (RMW), the volatile matter content was counted as the ratio of volatile matter weight ($DMW - RMW$) to the weight of the total dry matter [2,28]. The content of fixed carbon (FC) was obtained with the equation $FC$ (%) = $100 - VM$ (%) $- ash$ (%) [29].

The pH value of the six pecan biochars was measured with a PHS-3E pH meter (INESA, Shanghai, China). A DDS-307 conductivity meter from INESA was used to measure the EC value. In the experiment, the pecan biochar sample was mixed with deionized water at a 1:20 ratio ($w/v$), and then shaken for 24 h under the conditions of 25 °C and 200 rpm. Finally, the filtrate of the mixture through filter paper (qualitative filter paper, medium flow) was collected for the determination of the pH and EC.

An elemental analyzer (2400 II, PerkinElmer, Waltham, MA, USA) was used to test the elemental components (carbon—C, hydrogen—H, and nitrogen—N) of six pecan biochars. The oxygen (O) content was calculated using mass balance: $O$ (%) = $100 - (C + H + N)$ (%) $- ash$ (%). Based on the determined elemental contents, the atomic ratios of H/C, O/C, C/N and (O+N)/C were obtained. As for the determination of metal contents, the previously described method was referred to for digesting the biochar samples, and the digesting solution was used for the testing of metal contents by the iCAP 7400 ICP (Thermo Scientific, Waltham, MA, USA) [26].

### 2.4. Statistical Analyses

Excel 2010 was used to organize the data, and the values of the indexes were expressed as mean $\pm$ standard deviation. The plotting was performed using Origin 2022. SPSS 26.0 was used to conduct one-way ANOVA, and multiple comparisons were performed using the Duncan method, with the significance level set at $p < 0.05$ to ascertain the statistically significant differences between the physicochemical characteristics of the six pecan biochars.

## 3. Results and Discussion

### 3.1. Biochar Yield

As shown in Figure 1, the yields of the pecan biochars produced from the six types of feedstocks showed obvious differences. The yields of pecan biochars varied from high to low as follows: HB (45.9%), LB (41.7%), RB (39.5%), NSB (38.9%), BB (32.8%), and TB (32.1%). When compared with the yields of most previously reported biochars of plant biomasses, those of pecan biochars were relatively higher [17,28]. However, pecan biochars presented lower yields than those of the biochars made from wastewater sludge [30,31]. The differences in the biochar yields may be largely due to the different composition of the feedstocks and the pyrolytic conditions, such as the heating rate and pyrolysis temperature [32–34]. Notably, with reference to the previously described methods [26,27], the pecan biochars were prepared under a limited-oxygen atmosphere with no inert gas flow in this study, although various methods involved in the filling of inert gas have been reported to be performed in biochar preparation, such as the application of different types of inert gas [32,33,35], gas flow rates [30,31,36], and gas filling time [17,37,38]. As reported before, the changes in the reaction atmosphere could also impart different effects on the biochar yield or other properties [30–33]. For example, the corn stover biochars produced at 500 °C under pure $N_2$ and a mixture of 5% oxygen and 95% $N_2$ had the yields of 26.8% and 17.8%, respectively, with the yield difference of 9% [35]. The yields of Bambara groundnut (*Vigna subterranea*) biochars produced at 600 °C under $N_2$, $CO_2$, and $N_2/CO_2$ mixed gases (volume ratio of 3:1) varied in a range of 35%–40%, with relatively small differences [32].

Therefore, certain attention should be paid to the effects of pyrolysis atmosphere on biochar yield or other properties, apart from the concern of the impacts of feedstock type and other pyrolysis procedures.

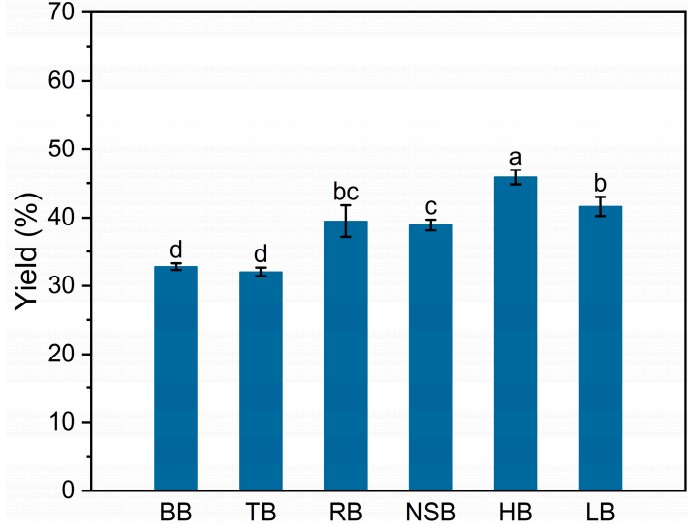

**Figure 1.** The yield of the biochars produced from six types of pecan feedstocks. The Duncan method was used for multiple comparisons, with the significant differences between six pecan biochars represented by different letters ($p < 0.05$). BB, Branch biochar; TB, Trunk biochar; RB, Root biochar; NSB, Nut shell biochar; HB, Husk biochar; LB, Leaf biochar.

Biochar is composed of ash and incompletely pyrolyzed biomass, and the yield of the biochar has a close relation to the contents of ash and incompletely pyrolyzed biomass. Because lignin was harder to be decomposed than other constituents of the biomass, with a relatively wide range of temperatures (300–700 °C), it is an important component of residual biomass after pyrolysis at 500 °C [39,40]. In this study, no significant correlation was found between yield and ash according to the correlation analysis. Therefore, it was speculated that the pecan biochar yield may be mainly affected by the content of incompletely pyrolyzed biomass, such as lignin.

### 3.2. Particle Size Distribution Characteristics, Surface Pore Property, and SEM Analysis

According to the analysis of biochar particles, the particles of six pecan biochars were distributed in a similar size range, which was between 1.65 μm and 549.54 μm (Figure S1). However, there were differences in the distribution characteristics of pecan biochar particles, with the particles of the TB, RB, HB, LB, BB, and NSB distributed in 17.38 μm, 19.95 μm, 34.67 μm, 52.48 μm, 69.18 μm, and 69.18 μm, with the highest volume percentage, respectively. As was shown in Table 1, the SSA, TPV, and APD of pecan biochars were 1.36–141.15 m$^2$/g, 0.0118–0.0994 cm$^3$/g, and 2.82–34.58 nm, respectively. Among those of the six pecan biochars, the TB and RB exhibited a larger SSA (141.15 and 75.32 m$^2$/g, respectively) and TPV (0.0994 and 0.0677 cm$^3$/g, respectively), while the APD of them was smallest (2.82 and 3.60 nm, respectively). In contrast with other previously reported biochars, such as maize stack, wheat straw (*Triticum aestivum*), pig mantle, and sludge, the TB and RB also had a better performance in terms of the SSA [28,41]. However, the SSA of the NSB obtained in this study (1.36 m$^2$/g) appeared smaller than the reported value of NSB in other studies [38,42]. As reported by Yang et al. (2023),the SSA depended on the size of the biochar particles, and the biochar particle size was negatively correlated with the SSA [42]. Therefore, apart from the differences in the method of material treatment, preparation procedure, etc., the possible differences in biochar particle size (not enough information was provided to compare the particle size of NSB in different studies) may be an important factor that led to the smaller SSA of NSB obtained in this study. As for the differences in the SSA and the pore characteristics shown among the six pecan biochars, it

can be found that the size distribution of the biochar particles may be an important factor. This is evident from the results indicating that the particles of TB and RB were distributed at smaller sizes with the highest volume percentage, while the highest percentage of NSB particles were found at larger sizes. Additionally, these differences can be attributed to the effects of the feedstocks with potentially distinct organic structures [13,14,28]. Therefore, the effect of the biochar particle size on the SSA and pore characteristics should be given more attention in further studies on biochars and their applications outside of the main factors (feedstock type, preparation procedure, etc.) that affect biochar properties. As revealed in many studies, the SSA and TPV of biochar are usually positively related to pollutant adsorption, and the biochar with a larger SSA and porosity also has a great potential to promote the improvement of the soil structure and their water-holding ability [43,44]. Given the results from this study, the application of the TB and RB with larger SSAs and potentially more adsorption sites may be more conducive to soil amendment and pollutant adsorption.

**Table 1.** The surface pore properties of the six pecan biochars.

| Biochar | Specific Surface Area | Total Pore Volume | Average Pore Diameter |
|:---:|:---:|:---:|:---:|
| | $m^2\ g^{-1}$ | $cm^3\ g^{-1}$ | nm |
| BB | 3.79 | 0.0160 | 16.85 |
| TB | 141.15 | 0.0994 | 2.82 |
| RB | 75.32 | 0.0677 | 3.60 |
| NSB | 1.36 | 0.0118 | 34.58 |
| HB | 5.83 | 0.0128 | 8.79 |
| LB | 2.78 | 0.0224 | 32.21 |

Note: BB, Branch biochar; TB, Trunk biochar; RB, Root biochar; NSB, Nut shell biochar; HB, Husk biochar; LB, Leaf biochar.

The surface morphology of the pecan biochars had obvious pore structures, as shown in Figure 2, and most of them were irregular in shape, which is consistent with most previous studies [29,42,45,46]. The formation of the pore structure was mainly attributed to the breakdown of the biomass components and the progressive volatilization of organic matter during pyrolysis [47]. In addition, the obvious differences can be observed in the morphological properties of the six pecan biochars. Overall, the BB, TB, and RB presented tubular shapes with elliptical pores, while the NSB, HB, and LB exhibited flaky or cluster structures, with relatively rougher outer surfaces and irregular pores.

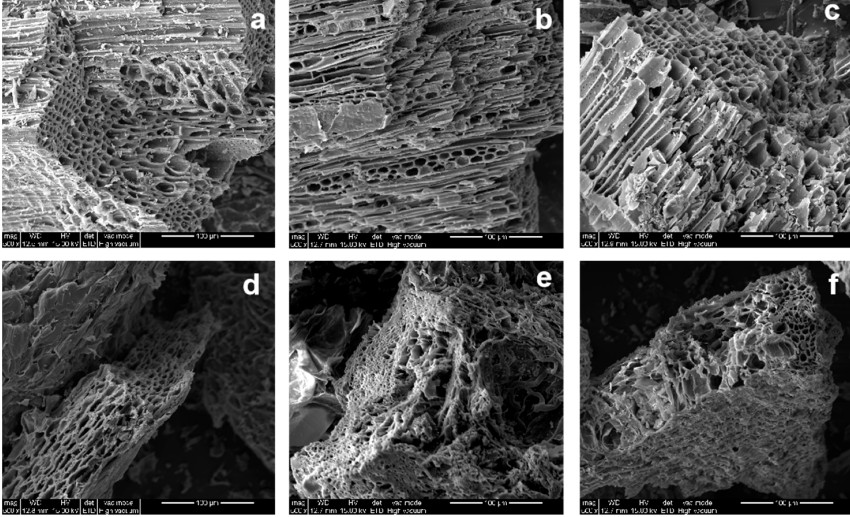

**Figure 2.** Scanning electron microscope images of the six pecan biochars. (**a**) BB, Branch biochar; (**b**) TB, Trunk biochar; (**c**) RB, Root biochar; (**d**) NSB, Nut shell biochar; (**e**) HB, Husk biochar; (**f**) LB, Leaf biochar.

### 3.3. FTIR Spectroscopy Analysis

Figure 3 showed that the six pecan biochars were mainly dominated by aromatic rings, and they had main peaks similar to those of the aromatic, aliphatic, and phenolic groups. A strong absorption peak was shown near 3400 cm$^{-1}$, and its occurrence was ascribed to the stretching vibration mode of [O-H] from carboxylic acids, alcohols, and phenols. The peaks at 2925 and 2855 cm$^{-1}$ represented the [C-H] vibration in the aromatic structure [31]. In addition, the stretching vibrations of [C=O] or [C=C] were also observed near 1600 cm$^{-1}$ [28]. The peak intensities at 1380 cm$^{-1}$ may be attributable to saturated [C-H] bending vibrations [41]. The peak near 1108 cm$^{-1}$ was possibly generated by [C-O] and [O-H] vibrations, which may be characteristic absorption peaks of cellulose and hemicellulose in incompletely pyrolyzed biomass [48]. Even though the types of surface functional groups presented no obvious differences among the six pecan biochars, variations in peak intensities were found at certain wavenumbers. For example, the peak intensities at 2925 and 2855 cm$^{-1}$ were obviously higher in the BB than in the other biochars. In the study on the properties of maize stalk, *Pinus*, and *Lantana camara* biochars, absorption peaks also appear near 2900 cm$^{-1}$ in the pine needle biochar, but this was not present in the maize stalk and *lantana camara* biochars [28]. Therefore, it was speculated that different feedstock types may cause differences in the absorption peak intensities or the types of specific functional groups.

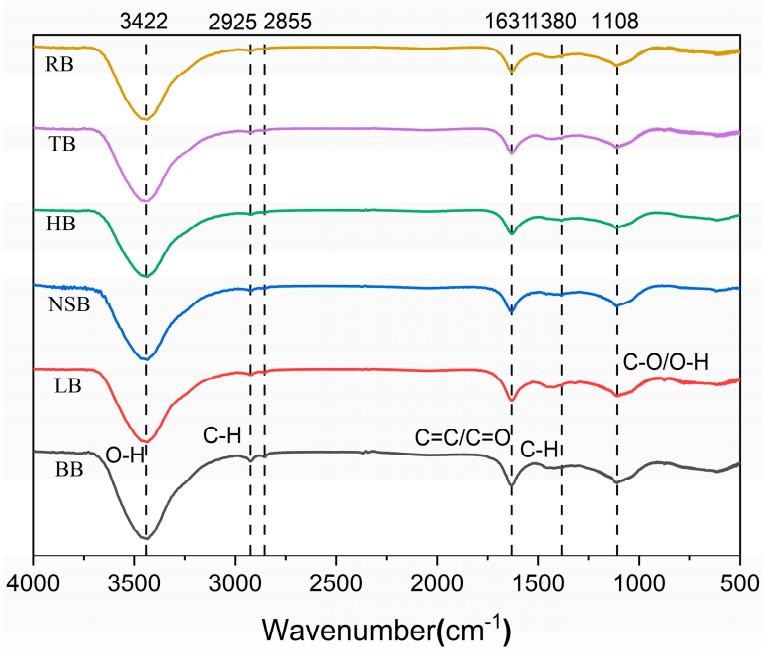

**Figure 3.** The FTIR spectrum of the six pecan biochars. FTIR: fourier transform infrared; BB, Branch biochar; TB, Trunk biochar; RB, Root biochar; NSB, Nut shell biochar; HB, Husk biochar; LB, Leaf biochar.

### 3.4. Proximate Analysis

Ash refers to the residue from the pyrolysis of biomass and is mainly composed of inorganic minerals. According to previous reports, biochars usually contain 0.3%–92.4% of ash [49]. As shown in Figure 4, the ash contents of the six pecan biochars were 7.8%–18.4%, which were higher than those of other woody plants, but far lower than those of the biochars from other biomasses, such as mud, poultry waste, and rice straw (*Oryza sativa*) [17,31,50–52]. Among the ash contents of the six biochars, those of the RB (18.4%) and LB (18.1%) were the highest, which may be attributed to the high inorganic mineral components contained in their raw materials [14,53]. The ash content of the NSB (7.8%) was the lowest, which was consistent with the previously reported ash content (5.0%) of the pecan shell biochar prepared at 450 °C [38].

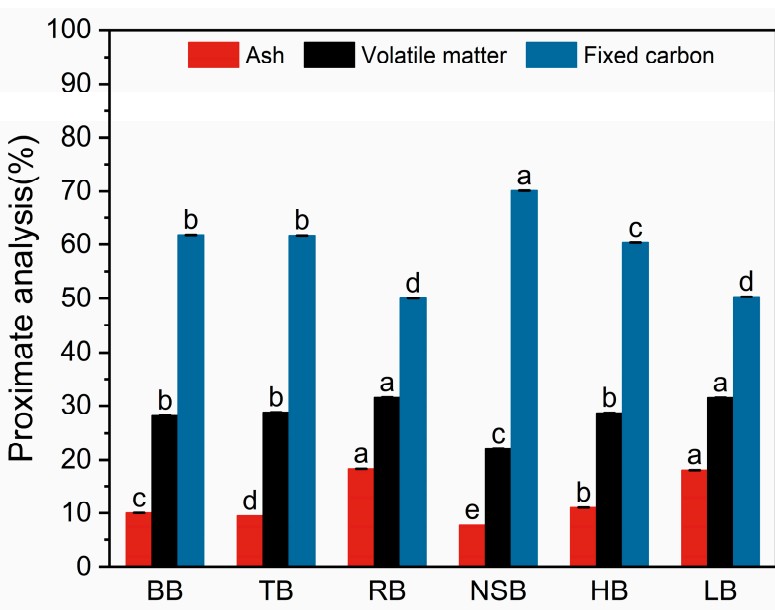

**Figure 4.** The proximate analysis of the six pecan biochars. The Duncan method was used for multiple comparisons, with the significant differences between six pecan biochars represented by different letters ($p < 0.05$). BB, Branch biochar; TB, Trunk biochar; RB, Root biochar; NSB, Nut shell biochar; HB, Husk biochar; LB, Leaf biochar.

The volatile matter (VM) contents of pecan biochars were 22.1%–31.6%. The LB (31.5%) and RB (31.6%) presented significantly higher VM contents than those of the other pecan biochars ($p < 0.05$), while the VM content of the NSB was the lowest. Compared with those of the rice straw (*Oryza sativa*), sludge, and wheat straw biochars, the LB and RB also presented higher VM contents [28,30,54]. However, the VM contents of pecan biochars were relatively lower than those of woody plant biochars, such as oak and pine [17]. As well as the ash content and the volatile matter, fixed carbon (FC) is an important part of biochars, which represents the remaining carbon in the solid structure of biochar. The fixed carbon in biochar is recalcitrant in nature compared to volatile matter, and its microbial degradation in soil can last for even thousands of years [55]. Herein, the FC contents of pecan biochars were 50.1%–70.1%, which were higher than those of other biochars, such as farmyard manure, rice husks, and sludge samples [14,17,30,31]. Among the FC of all pecan biochars, that of the NSB were the highest, whereas those of the RB and LB were significantly lower at 50.1% and 50.3%, respectively. Based on the correlation analysis, the ash contents were found negatively correlated with fixed carbon, which was consistent with previous reports [14,56]. This relationship may be due to the inhibitory role of ash in the formation of aromatic structures [14,56,57].

*3.5. pH and EC Analysis*

The pH value of the biochar is an important reference index in soil amendments. As shown in Figure 5, six pecan biochars presented the pH values of 7.90–9.87, appearing alkaline, which was in accordance with the results of most studies [8]. However, it has also been suggested in some studies that biochar can be acidic when subjected to low pyrolysis temperatures [26]. The difference in the acidity and alkalinity of biochar may be related to the variations in inorganic components and surface functional groups during the pyrolysis process [58]. It was found from the comparison of pH values in different pecan biochars that the HB (9.87), LB (8.60), and NSB (8.52) had substantially greater pH values than those of the other biochars ($p < 0.05$), whereas the RB had the lowest pH value (7.90). It has been reported that the pH values of acidic soils tended to increase after the application of the biochar with alkaline, and its neutralizing effect on acidic soils increased with the alkalinity of the biochar, which will also lead to an increase in the utilization efficiency of

soil nutrients [59]. It can thus be speculated that those alkaline pecan biochars, especially the more highly alkaline biochars (such as HB), may generally have a better performance in the amelioration of acidic soils. Certainly, alkaline biochars can also reduce the pH value of saline-alkali soils, but the effectiveness of its application effect was also impacted by certain factors, such as soil type and application rate [60].

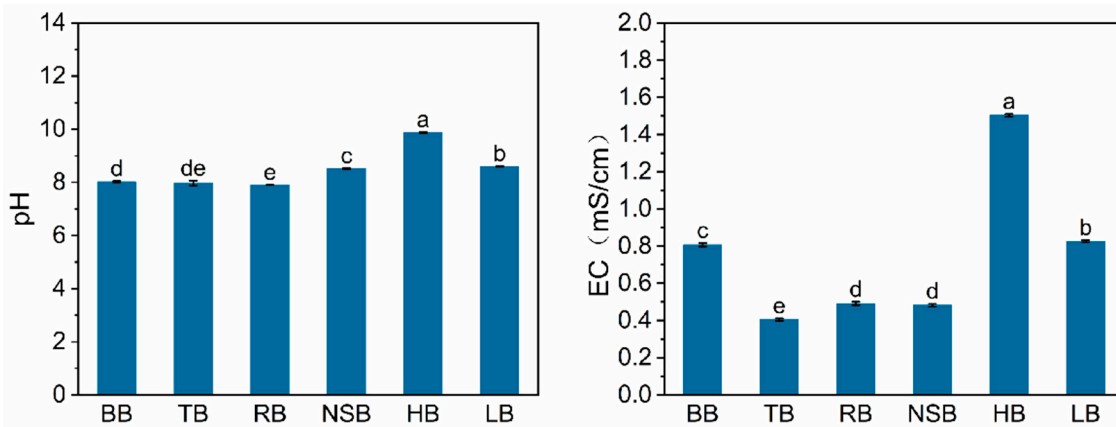

**Figure 5.** The pH and EC values of the six pecan biochars. The Duncan method was used for multiple comparisons, with the significant differences between six pecan biochars represented by different letters ($p < 0.05$). EC: electrical conductivity; BB, Branch biochar; TB, Trunk biochar; RB, Root biochar; NSB, Nut shell biochar; HB, Husk biochar; LB, Leaf biochar.

Electrical conductivity (EC) refers to the content of total soluble cations. The findings revealed that the six pecan biochars had EC values of 0.41–1.50 mS/cm, which presented lower levels when compared with those of the biochars made from other biomasses, such as wheat straw, corn straw, rice straw, and *Jatropha curcas* [16,36]. Among the pecan biochars, the HB had the highest EC value, followed by the LB (0.83 mS/cm) and BB (0.81 mS/cm). This was consistent with the size order in the EC values of the *Jatropha curcas* leaf, branch, and trunk biochar [16]. As reported, the biochar with a high content of soluble cations tended to have an adverse effect on soil properties and plant growth [50]. Therefore, pecan biochars, especially TBs, NSBs, and RBs with lower EC values, may offer a greater safety profile in soil amendment.

*3.6. Elemental Analysis*

From the elemental analysis, the C content was the highest in the six pecan biochars with the contents of 52.2%−74.1%, and the O, H, and N contents were 4.2%–33.6%, 2.7%–2.9%, and 0.9%–2.0%, respectively (Table 2). When compared to the biochars from oak and pine feedstocks, pecan biochars had lower C contents, but relatively higher N contents [17]. C is the main component of biochar, and biochars with a high C content could increase the level of soil organic carbon. In this study, the C contents of RBs, NSBs, and LBs exceeded 70%, which were significantly higher than those of most reported biochars, and those of the NSB were similar to those of the previously reported pecan shell biochars [16,31,38,61]. Therefore, when compared to other biochars, RBs, NSBs, and LBs may be more beneficial for improving soil organic matter content [62,63]. The C/N ratio is a key factor that can affect the soil nutrient cycle and metabolic processes [64]. According to previous reports, biochars with an excessive C/N ratio tended to inhibit soil microbial decomposition and decrease the available nitrogen in the soil [65,66]. Herein, the C/N ratios of HBs (31.54) and TBs (31.40) were the lowest values among those of all pecan biochars. Accordingly, it may be more beneficial for soil nutrient cycling to apply these biochars to the soil.

**Table 2.** Elemental analysis of the six pecan biochars.

| Biochar | C(%) | H(%) | N(%) | O(%) | H/C | O/C | C/N | (O + N)/C |
|---------|------|------|------|------|-----|-----|-----|-----------|
| BB | 66.0 | 2.8 | 1.4 | 19.9 | 0.50 | 0.23 | 54.57 | 0.24 |
| TB | 52.2 | 2.7 | 1.9 | 33.6 | 0.63 | 0.48 | 31.40 | 0.51 |
| RB | 71.1 | 2.7 | 2.0 | 5.9 | 0.45 | 0.06 | 42.30 | 0.09 |
| NSB | 73.5 | 2.8 | 1.7 | 14.7 | 0.46 | 0.15 | 51.04 | 0.17 |
| HB | 53.3 | 2.9 | 2.0 | 30.9 | 0.65 | 0.43 | 31.54 | 0.47 |
| LB | 74.1 | 2.8 | 0.9 | 4.2 | 0.45 | 0.04 | 101.66 | 0.05 |

Note: BB, Branch biochar; TB, Trunk biochar; RB, Root biochar; NSB, Nut shell biochar; HB, Husk biochar; LB, Leaf biochar.

Biochar contains diverse mineral elements, and the application of biochar will increase the nutrient level in soil [67]. Herein, it was found that the Ca contents were generally higher than those of other nutrients in different pecan biochars. Among those of all the pecan biochars, the HB contained the highest contents of N, K, and Mg, and the P and Ca contents were highest in NSBs and LBs, respectively. However, the levels of the most commonly tested nutrients (i.e., P, K, Mg) were lowest in TBs (Table 3). In the study on biochars from the leaves, the trunks and branches of *Jatropha curcas*, the N, P, K, and Mg contents were highest in the leaf biochar, whereas the trunk biochar presented the highest Ca content [16]. Additionally, according to a previous study on the biochars from pecan branches, leaves, and nutshells, the N, Ca, and Mg contents were highest in leaf biochars pyrolyzed at 500 °C, with the highest P and K levels present in branch biochars [26]. In terms of the nutrient contents for the BB, TB, LB, and NSB obtained in the present study, the LB contained the lowest N level, but the highest Ca and Mg contents; the P and K contents were the highest in the NSB, with the P content in the LB and the K content in the BB second only to those of the NSB. Through comparison, it was suggested that there were some common trends in the nutrient contents in different biochars, such as Mg content being highest in the leaf biochar, but the changes in the distribution characteristics of nutrients was also found in different biochars. Therefore, it can be speculated that the variable results on the nutrient contents in the various biochars may be attributed to plants having potentially distinct elemental compositions across different tissues (or organs), or the exposure to different management levels, feedstock origins, times for feedstock collection, etc. Notably, based on the International Biochar Initiative (IBI, 2014), the Zn, Cr, Cu, Pb, and Cd contents in the six pecan biochars were all in the safety range (the maximum allowed thresholds for them were 200–7000, 64–1200, 63–1500, 70–500, and 1.4–39 mg kg$^{-1}$, respectively). Therefore, there would be no potential risk of heavy metal pollution in the six pecan biochars to the environment after the application of the biochars (Table S1) [29,68].

**Table 3.** Contents of phosphate and metal elements in the six pecan biochars.

| Biochar | P g kg$^{-1}$ | K | Ca | Mg |
|---------|------|------|------|------|
| BB | 2.54 ± 0.01 b | 14.40 ± 0.27 c | 22.84 ± 0.81 b | 3.52 ± 0.28 bc |
| TB | 0.63 ± 0.00 e | 2.65 ± 0.12 f | 15.98 ± 0.02 c | 2.68 ± 0.03 d |
| RB | 1.26 ± 0.01 d | 5.47 ± 0.15 e | 14.27 ± 0.62 d | 3.71 ± 0.15 b |
| NSB | 4.15 ± 0.04 a | 15.23 ± 0.20 b | 9.75 ± 0.16 f | 3.09 ± 0.25 cd |
| HB | 1.37 ± 0.01 c | 22.35 ± 0.06 a | 11.75 ± 0.15 e | 5.56 ± 0.15 a |
| LB | 2.56 ± 0.05 b | 8.93 ± 0.08 d | 31.51 ± 0.90 a | 5.41 0.49 a |

Note: Duncan method was used for multiple comparisons, with the significant differences between six pecan biochars represented by different letters ($p < 0.05$). BB, Branch biochar; TB, Trunk biochar; RB, Root biochar; NSB, Nut shell biochar; HB, Husk biochar; LB, Leaf biochar.

The Van Kresen diagram was initially developed as a two-dimensional model diagram using the H/C and O/C ratios, and it reflects the evolution of coal or coal-forming organic matter. It has recently been applied more frequently to evaluate the aromaticity and maturity in biochar structures [17,69]. In the present study, the H/C ratios of the six pecan

biochars were ranging from 0.45 to 0.65 (Figure 6). The H/C of most well-pyrolyzed organic feedstocks should be less than 0.7, as regulated by the European Biochar Certificate (EBC, Arbaz, Switzerland) [70]. Therefore, all six pecan biochars in this study met this guideline. In addition, as reported previously, the biochars of O/C ratios ranging from 0.2 to 0.6 potentially had a half-life of 100–1000 years, and, when the O/C ratios of biochar were smaller than 0.2, their half-life would exceed 1000 years [55,71]. From the Van Kraveren diagram, the O/C ratios of the LB, RB, NSB, BB, HB, and TB were 0.04, 0.06, 0.15, 0.23, 0.43, and 0.48, respectively. Among all pecan biochars, the O/C ratios of the LB, RB, and NSB were not more than 0.2, which indicated that they were highly carbonized and had high a stability and capacity for carbon sequestration.

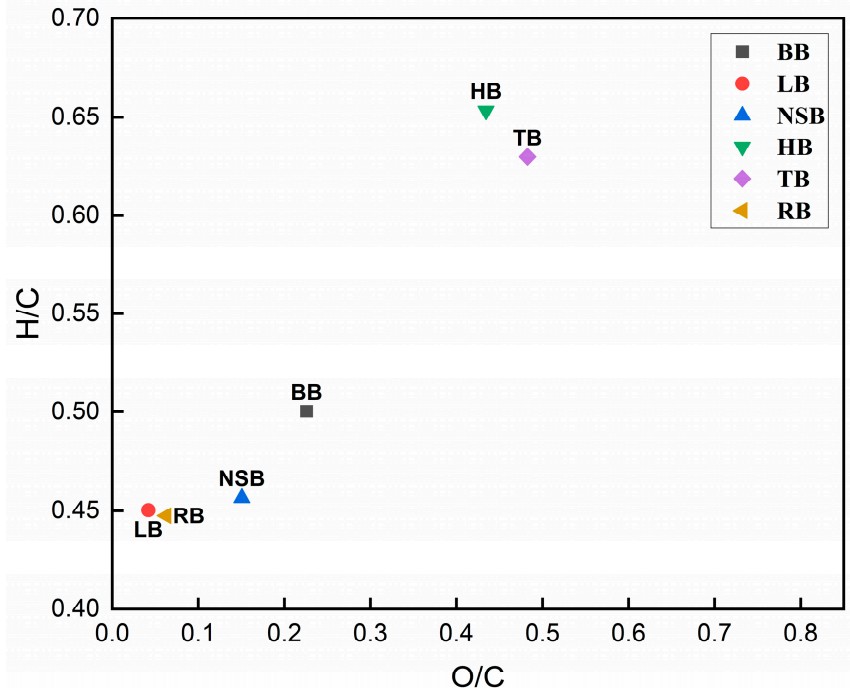

**Figure 6.** Van Krevelen plot of the six pecan biochars. BB, Branch biochar; TB, Trunk biochar; RB, Root biochar; NSB, Nut shell biochar; HB, Husk biochar; LB, Leaf biochar.

*3.7. Correlation Analysis*

In Table 4, the results of the correlation analysis were shown on the physiochemical properties of the six pecan biochars. Based on the results, the biochar yield showed positive correlations with the pH value and Mg content. However, a negative correlation was previously reported between the biochar yield and pH value, which may result from the different properties of raw materials and preparation procedures of biochars [16,36]. The TPV was found to be positively correlated with the SSA but presented a negative correlation with the K content, and the APD was positively related to the P content. The FC content was negatively correlated with ash and the VM contents, while the ash content presented a positive relation with the VM content. The EC and pH value showed a positive relation, with a negative relation between the N and Ca contents, which has also been reported before [16]. In addition, the C content was positively related to the O content and the H/C, O/C, and (O + N)/C ratios, whereas the O content showed a negative correlation with the three elemental ratios; this was consistent with the previous reports [36]. The Ca content also demonstrated positive correlation with the C/N ratio.

**Table 4.** Correlation analysis of the physicochemical characteristics for the six pecan biochars (Pearson method).

| | Yield | SSA | TPV | APD | Ash | VM | FC | pH | EC | C | H | N | O | H/C | O/C | C/N | (0 + N)/C | P | K | Ca |
|---|---|---|---|---|---|---|---|---|---|---|---|---|---|---|---|---|---|---|---|---|
| SSA | −0.52 | | | | | | | | | | | | | | | | | | | |
| TPV | −0.51 | 0.99 ** | | | | | | | | | | | | | | | | | | |
| APD | 0.20 | −0.71 | −0.68 | | | | | | | | | | | | | | | | | |
| Ash | 0.36 | 0.01 | 0.14 | −0.06 | | | | | | | | | | | | | | | | |
| VM | 0.12 | 0.28 | 0.37 | −0.47 | 0.82 * | | | | | | | | | | | | | | | |
| FC | −0.27 | −0.13 | −0.25 | 0.25 | −0.97 ** | −0.94 ** | | | | | | | | | | | | | | |
| pH | 0.81 * | −0.51 | −0.56 | 0.12 | −0.14 | −0.11 | 0.13 | | | | | | | | | | | | | |
| EC | 0.68 | −0.54 | −0.58 | −0.05 | −0.01 | 0.16 | −0.07 | 0.89 * | | | | | | | | | | | | |
| C | 0.11 | −0.48 | −0.38 | 0.69 | 0.45 | −0.07 | −0.24 | −0.34 | −0.37 | | | | | | | | | | | |
| H | 0.55 | −0.50 | −0.58 | 0.23 | −0.40 | −0.27 | 0.36 | 0.92 ** | 0.83 * | −0.43 | | | | | | | | | | |
| N | −0.03 | 0.52 | 0.45 | −0.73 | −0.35 | −0.18 | 0.29 | 0.08 | −0.02 | −0.59 | −0.04 | | | | | | | | | |
| O | −0.23 | 0.36 | 0.24 | −0.50 | −0.72 | −0.24 | 0.54 | 0.31 | 0.29 | −0.94 ** | 0.49 | 0.57 | | | | | | | | |
| H/C | 0.03 | 0.36 | 0.26 | −0.58 | −0.46 | 0.02 | 0.27 | 0.49 | 0.48 | −0.99 ** | 0.58 | 0.54 | 0.94 ** | | | | | | | |
| O/C | −0.18 | 0.42 | 0.31 | −0.55 | −0.63 | −0.14 | 0.44 | 0.33 | 0.31 | −0.97 ** | 0.48 | 0.57 | 0.99 ** | 0.97 ** | | | | | | |
| C/N | 0.17 | −0.48 | −0.39 | 0.72 | 0.52 | 0.26 | −0.43 | −0.08 | −0.03 | 0.69 | −0.04 | −0.97 ** | −0.71 | −0.63 | −0.70 | | | | | |
| (0 + N)/C | −0.18 | 0.43 | −0.32 | −0.57 | −0.63 | −0.14 | 0.44 | 0.33 | 0.30 | −0.98 ** | 0.47 | 0.59 | 0.99 ** | 0.97 ** | 1.00 ** | −0.71 | | | | |
| P | 0.09 | −0.72 | −0.72 | 0.91 * | −0.26 | −0.69 | 0.47 | 0.03 | −0.12 | 0.70 | 0.13 | −0.48 | −0.44 | −0.62 | −0.53 | 0.46 | −0.54 | | | |
| K | 0.57 | −0.78 | −0.84 * | 0.28 | −0.39 | −0.42 | 0.42 | 0.80 | 0.80 | −0.12 | 0.81 | 0.03 | 0.23 | 0.24 | 0.17 | −0.11 | 0.17 | 0.39 | | |
| Ca | −0.12 | −0.21 | −0.12 | 0.33 | 0.52 | 0.56 | −0.56 | −0.21 | 0.03 | 0.34 | −0.11 | −0.88 * | −0.43 | −0.34 | −0.40 | 0.85 * | 0.43 | 0.04 | −0.27 | |
| Mg | 0.83 * | −0.56 | −0.53 | 0.18 | 0.48 | 0.44 | −0.49 | 0.75 | 0.82 * | 0.03 | 0.60 | −0.38 | −0.19 | 0.09 | −0.14 | 0.43 | −0.05 | 0.50 | 0.37 | |

Note: The values in the figure are the correlation coefficients. "*" refers to a correlation at significant level ($p < 0.05$), while "**" represents a correlation at extremely significant level ($p < 0.01$).

### 3.8. Principal Component Analysis

From the principal component analysis (PCA), the first two principal components had the variance contributing rates of 39.8% and 29.9%, respectively (Figure 7). The N and O contents and the H/C, O/C, and (O + N)/C ratios were found with high positive loadings in PC1, while the C content and the C/N ratio presented dominantly negative loadings, indicating that the contents of the main constituent elements and their atomic ratios were the main contributor to PC1. Meanwhile, the HB and TB were observed to be distributed in quadrant I and IV, respectively, with high positive scores in PC1, whereas the high negative score was for the LB. This illustrated that the HB and TB possessed excellent levels in the contents of C, O, and N, and the H/C, O/C, and (O + N)/C ratios, with high C content and C/N ratio in the LB, which was consistent with the results of elemental analysis. Additionally, it was obtained that the pH and EC values and the K and H contents were the main positive contributors to PC2, and the HB with the high positive scores in PC2 was implied to have high values in these indexes. In contrast, the SSA and TPV exhibited the high negative relations with PC2, and the TB and RB showed a high negative score in PC2, suggesting that the TB and RB had better performances in the SSA and TPV. According to the distribution of the pecan biochars in the coordinates of PC1 and PC2, it was found that the BB, LB, and NSB were clustered in quadrant II with close relationships, and they appeared far from the HB, RB, and TB that were located in quadrants I, III, and IV, respectively. This result was different from that obtained from *Jatropha curcas* biochars where the biochars from trunk and branch presented a close relationship, and they are far from leaf biochars in comparison, which may be attributed to the differences in plant characteristics [16].

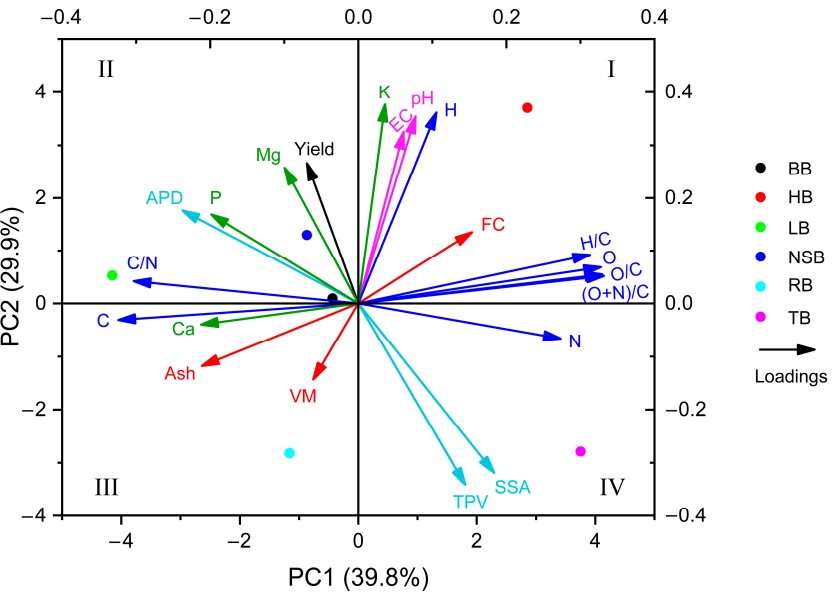

**Figure 7.** PCA biplot of the physicochemical parameters for the six pecan biochars. PCA: principal component analysis; BB, Branch biochar; TB, Trunk biochar; RB, Root biochar; NSB, Nut shell biochar; HB, Husk biochar; LB, Leaf biochar.

### 4. Conclusions

This study revealed that different types of feedstocks induced significant changes in biochar properties and structures during pyrolysis. From FTIR analysis, the types of surface functional groups were all similar, but there were some differences in the peak intensity at certain wavenumbers, such as the higher peak intensity observed in the BB near 2900 cm$^{-1}$. The TB and RB possessed the highest SSA, and they may be more conducive to soil amendment and pollutant adsorption. The HB, LB, and NSB had significantly higher alkalinity, so they may have a better performance in increasing the pH of acidic soils. The HB had the highest contents of N, K, and Mg, with higher P and Ca levels in

the NSB and LB, respectively, suggesting their greater potential in the supplementation of mineral nutrition after field application. The LB, RB, and NSB were found to contain higher contents of C (over 70%) with lower O/C (no more than 0.2), and they may be more beneficial for organic matter improvement and carbon sequestration. Based on these results, pecan biochars can be selected as suitable materials for applications in agriculture and environmental fields.

**Supplementary Materials:** The following supporting information can be downloaded at: https://www.mdpi.com/article/10.3390/f15020366/s1, Figure S1: Size distribution of the six pecan biochars; Table S1: Heavy metal contents in the six pecan biochars.

**Author Contributions:** Conceptualization, M.Z., F.P., J.Y. and Z.L.; methodology, M.Z. and Z.L.; sample collection, investigation and data curation, M.Z.; formal analysis, M.Z. and Z.L.; writing—original draft preparation, M.Z.; writing—review and editing, M.Z., F.P., J.Y. and Z.L. All authors have read and agreed to the published version of the manuscript.

**Funding:** This research was funded by the National Key Research and Development Project of China (2021YFD1000403), the Central Financial Services Demonstration Funds for Forestry Science and Technology Promotion Project of China ((2022)G04), the Postgraduate Research and Practice Innovation Program of Jiangsu Province (KYCX21_0911), the Priority Academic Program Development of Jiangsu Higher Education Institutions (PAPD), the Special Fund on Technology Innovation of Carbon Dioxide Peaking and Carbon Neutrality of Jiangsu Province (BE2022306), the Scientific Research Foundation for Talented Scholars of Institute of Botany, Jiangsu Province and Chinese Academy of Sciences (JIBTF202209).

**Data Availability Statement:** The data are not publicly available due to privacy restrictions.

**Acknowledgments:** We greatly thank the editors and anonymous reviewers for their valuable suggestions on our manuscript.

**Conflicts of Interest:** The authors declare no conflicts of interest.

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
