# Peer review of "Feedstock-Induced Changes in the Physicochemical Characteristics of Biochars Produced from Different Types of Pecan Wastes"

_forests, doi:10.3390/f15020366_

Round 1

Reviewer 1 Report (Previous Reviewer 1)

Comments and Suggestions for Authors

The current manuscript evaluates the biochar based on 500C pyrolysis temperature only which is not enough to be published unless you test it as a soil amendment or adsorption experiment. Moreover, what is the percentage of root and trunk to be used in biochar production??? what is the new merit between this submitted manuscript and the previously published paper by the corresponding author???

https://www.sciencedirect.com/science/article/pii/S092666902300403X

Author Response

Reviewer 2 Report (Previous Reviewer 2)

Comments and Suggestions for Authors

I consider that the article has been significantly improved, although, from my point of view, the biochar production process does not precisely align with pyrolysis. The absence of an inert gas displacing oxygen in the carbonization process could be linked to the traditional small-scale preparation of biochar in containers, but not with the current pyrolysis processes commonly used and described in the literature. This aspect could be discussed, attempting to establish connections with properties of both types of charcoal.

Author Response

Reviewer 3 Report (Previous Reviewer 3)

Comments and Suggestions for Authors

The authors have addressed all the comments and additional analysis suggested by the reviewer during the previous submission of this manuscript. Thus the manuscript may be considered for publication in journal of Forests.

Author Response

Thank you for reviewing our manuscript again and thanks for your approval of our revised manuscript. 

Reviewer 4 Report (Previous Reviewer 4)

Comments and Suggestions for Authors

The authors answered all my questions of first round of review of this manuscript (forests- 2715645). It has been significantly improved. 

Author Response

Thank you for reviewing our manuscript again and thanks for your approval of our revised manuscript.

Round 2

Reviewer 1 Report (Previous Reviewer 1)

Comments and Suggestions for Authors

The manuscript has 38% similarity index based on Turnitin. Moreover, the submitted manuscript is basically a part of the previously published paper by the corresponding author entitled "Comparative analysis of the properties of biochars produced from different pecan feedstocks and pyrolysis temperatures"
https://www.sciencedirect.com/science/article/pii/S092666902300403X

Moreover, no means to use Pecan root and trunk to produce biochar as the amount is not enough and will not cut a tree to produce biochar!!!!

Author Response

This manuscript is a resubmission of an earlier submission. The following is a list of the peer review reports and author responses from that submission.

Round 1

Reviewer 1 Report

Comments and Suggestions for Authors

The current manuscript evaluates the biochar based on 500C pyrolysis temperature only which is not enough to be published unless you test it as a soil amendment or adsorption experiment. Moreover, what is the percentage of root and trunk to be used in biochar production??? what is the new merit between this submitted manuscript and the previously published paper by the corresponding author???

https://www.sciencedirect.com/science/article/pii/S092666902300403X

Reviewer 2 Report

Comments and Suggestions for Authors

The article is well-written, and the methodology is very clear, although I would like to make some comments with the aim of improving the manuscript. I believe that the keywords should be modified to exclude terms present in the title, thus facilitating the article's search. Secondly, I find that the description of the pyrolysis process is not adequately detailed; a muffle furnace is mentioned, but since it involves pyrolysis, it is necessary to specify the characteristics of a reactor, the inert gas forming the atmosphere, its flow rate, and residence time. Additionally, the study could benefit from a physical characterization of the obtained charcoal, detailing the particle size distribution and its potential phytotoxicity when applied as an amendment in different plant species. Despite these comments, the properties described in the manuscript are well-determined, and the methodology is well-explained.

Reviewer 3 Report

Comments and Suggestions for Authors

The authors have worked on the production and characterization of biochar produced from pecan wastes. The manuscript is well-written with appropriate methodologies for differentiation in characteristics of the biochar depending on its source. But there are some gaps in the study, which has to be addressed by the authors for publication consideration. 

1. The Biochar produced by the authors shows porous structure in SEM analysis. The porosity of the different Biochar's should be analyzed through BET analysis and can be compared to give new insights. This will improve the merit of publication.

Reviewer 4 Report

Comments and Suggestions for Authors

This paper may be published in Forests journal. It presents the results of biochar production from various plant raw materials of pecan (branches, trunks, roots, nutshells, husks, and leaves) under pyrolysis. It’s a systematic and sequential study. Some revision is required:

Major remarks:

1. The authors described how the different samples of pecan raw materials were collected, but did not specify the season of the year. How can the season and season weather influence the chemical composition of biochar? Some comments should be done.

2. Section 2.1. Why were such pyrolysis temperatures chosen? The papers on the production of boichar from pecan shells have published enough. How the characteristics of your samples match the literature data. Some comments should be done.

Minor remarks:

1. Line 78. The “EC“ abbreviation should be cleared when first mentioned .

2  Lines 144, 235. The “infrared spectrogram” should be replaced by “infrared spectrum”.

3 Lines 153-156. When determine the of volatile matter content, did the authors analyze the residue mass?

4 Lines 182-183 show the repeated information of Line 184.

5 Lines 192-194. The authors wrote that “Therefore, it was speculated that the pecan biochar yield may be affected by the content of incompletely pyrolyzed biomass, such as lignin.” On what basis is this conclusion based?

6 Lines 214. The authors wrote that “The types of functional groups of biochar can be qualitatively analyzed by FTIR”. Quantitative determination cannot be done by FTIR.

7 Lines 219-202. Why the functional groups of alcohols, phenols, and carboxylic acids do help to improve the alkalinity of biochar? Is it alkalinity?

8 Lines 249-242. The conclusion “The volatile matter was able to immediately increase soil organic carbon through enhanced microbial degradation, but it was easily broken down to CO2 and water in a short period. Therefore, there was a limited impact of volatile matter on the enhancement of soil organic matter in the long term. “ should be clarified.

9 Line 320 and then on to the text. I think that “N” should be replaced by “Na”.

10 Lines 369. “with pH and Mg” should be corrected by “with pH and Mg content”.

11 Lines 389-391. The authors wrote that “a close relation was also found between the biochars of branches and roots, both of which are mainly composed of xylem.” It should be clarified or corresponding references should be done.

12 Line 412. As authors wrote the vibration near 2900 cm-1 represents the vibration of [C-H] in the aromatic structure. C-H does not refer to functional groups. This phrase should be corrected.
